# Non-stationary coherent quantum many-body dynamics through dissipation

Berislav Buča [1], Joseph Tindall [1] & Dieter Jaksch [1,2]

The assumption that quantum systems relax to a stationary state in the long-time limit underpins statistical physics and much of our intuitive understanding of scientific phenomena. For isolated systems this follows from the eigenstate thermalization hypothesis. When an environment is present the expectation is that all of phase space is explored, eventually leading to stationarity. Notable exceptions are decoherence-free subspaces that have important implications for quantum technologies and have so far only been studied for systems with a few degrees of freedom. Here we identify simple and generic conditions for dissipation to prevent a quantum many-body system from ever reaching a stationary state. We go beyond dissipative quantum state engineering approaches towards controllable long-time non-stationarity typically associated with macroscopic complex systems. This coherent and oscillatory evolution constitutes a dissipative version of a quantum time crystal. We discuss the possibility of engineering such complex dynamics with fermionic ultracold atoms in optical lattices.

[1] Clarendon Laboratory, University of Oxford, Parks Road, Oxford OX1 3PU, UK. [2] Centre for Quantum Technologies, National University of Singapore, 117543 Singapore, Singapore. Correspondence and requests for materials should be addressed to B.B. (email: berislav.buca@physics.ox.ac.uk) or to J.T. (email: joseph.tindall@physics.ox.ac.uk) or to D.J. (email: dieter.jaksch@physics.ox.ac.uk)

The eigenstate thermalization hypothesis[1,2] (ETH) states that an isolated many-body quantum system with non-integrable Hamiltonian relaxes locally to a stationary equilibrium ensemble. For generic initial states local observables are given by thermal expectation values after a sufficiently long evolution time $t$. A generalized ETH holds if the system is integrable or under the influence of weak integrability breaking[1,2]. Equilibration occurs on relatively short timescales, typically within a few characteristic periods.

Perfect isolation is impossible in experiments and interactions with the environment will always provide additional relaxation mechanisms. The widely used—but notoriously difficult to prove—assumption of ergodicity states that even weak coupling to an environment enables the system to explore the entire connected nondecaying part of the system Hilbert space $\mathcal{H}$ as sketched in Fig. 1a. The evolution thus induces relaxation to a unique stationary state $\rho_\infty$ in the long-time limit.

In quantum technology platforms[3] a microscopic understanding of the environmental coupling allows control of the open system dynamics $\dot{\rho}(t) = \mathcal{L}\rho(t)$ of the density operator $\rho(t)$. The super-operator $\mathcal{L}$ can be engineered to possess a small number of controllable purely imaginary eigenvalues. The corresponding eigenstates are protected from the environment and form a decoherence-free subspace[4,5] where quantum information can be processed without leaking into the environment. Controlled dissipation can also lead to many-body pure states that are stationary[6–8].

The seemingly robust feature of relaxation to stationarity in quantum many-body systems presents a puzzle when contrasted with the emergence of nonstationary dynamics often observed in macroscopic systems[9,10]. Nonstationarity plays an important role in many areas ranging from microbiology[11,12] and neurobiological systems[9,13] across climate science[14,15] to financial time series[9,16]. It remains almost unstudied in quantum statistical physics where research of nonequilibrium setups mostly concentrates on currents of time-independent quantities. The question thus arises whether generic insights into the microscopic origins of nonstationary and complex long-time evolution may be gleaned from the study of highly controlled and well understood experiments in the quantum regime.

Here, we show that coupling to an environment can induce nonstationarity in many-body quantum systems that would otherwise relax, through mutual dephasing of its eigenstates, according to the ETH. Symmetry-preserving dissipation eliminates a large class of eigenstates and ensures constructive interference. It splits the nondecaying part of the Hilbert space into disjoint sectors schematically shown in Fig. 1b. In the long-time limit a dark Hamiltonian coherently drives the system between these disjoint parts leading to nondecaying oscillations in observables that are not entirely contained in one sector. We will give general conditions guaranteeing this situation and study an example realizable in current experiments with ultracold atoms.

## Results

**Conditions for nonstationarity in a many-body system.** Specifically, our starting point is the Lindblad master equation modeling a quantum system weakly coupled to an environment that acts as a source of noise. The main results presented here are also valid for open quantum systems beyond the Lindblad framework (see Supplementary Methods for details). The master equation is given by (setting $\hbar = 1$)

$$\dot{\rho}(t) = L\rho = -i[H, \rho(t)] + \sum_\mu \left( 2L_\mu \rho L_\mu^\dagger - L_\mu^\dagger L_\mu \rho - \rho L_\mu^\dagger L_\mu \right),$$

(1)

where the first term describes unitary evolution $i\dot{\rho} = [H, \rho(t)] = H\rho(t) - \rho(t)H$ of an isolated system with Hamiltonian $H$ and follows directly from the Schrödinger equation. The second term contains the jump operators $L_\mu$ arising from decoherence processes induced by the environment. Formally, the density operator will be nonstationary if the Liouvillian $\mathcal{L}$ has purely imaginary eigenvalues[17–19] $\mathcal{L}\rho_n = -i\mathfrak{H}\rho_n = -i\lambda_n \rho_n$ for eigen-operators $\rho_n$. Here, we have defined the dark Hamiltonian $\mathfrak{H}$ as the part of the evolution that is purely coherent.

The conceptually simplest situation where nonstationarity may occur is well understood for systems with few degrees of freedom[18–20]. All jump operators fulfill $L_\mu |\phi_n\rangle = 0$ for a subset of eigenstates $|\phi_n\rangle$ with eigenvalues $\omega_n$ of the Hamiltonian. These so-called dark states are perfectly decoupled from the environment and span a decoherence-free subspace[4]. Coherences between dark states evolve according to $\mathcal{L}|\phi_n\rangle\langle\phi_m| = i(\omega_m - \omega_n)|\phi_n\rangle\langle\phi_m|$ and undergo continued oscillations induced by the coherent part of the dynamics. The dark Hamiltonian $\mathfrak{H}$ may then be understood as a purge of unwanted eigenstates of the original Hamiltonian $H$. We give an example of a many-body dark Hamiltonian in the Supplementary Discussion.

A dark Hamiltonian is not required to be Hermitian and its eigenstates need not be pure. We concentrate on this more general and interesting case and show that it may lead to nonstationary and complex long-time dynamics. This case is realized if there exists an eigenoperator $A$ such that (see Supplementary Methods for details)

$$[H, A] = -\lambda A \text{ and } \left[L_\mu, A\right] = \left[L_\mu^\dagger, A\right] = 0 \ \forall \mu,$$

(2)

with real valued $\lambda$. We find $\mathcal{L}\rho_{nm} = i(m-n)\lambda \rho_{nm}$ for operators $\rho_{nm} = A^n \rho_\infty \left(A^\dagger\right)^m$ and integer $m, n > 0$. Here, $\rho_\infty$ is a stationary state with $\mathcal{L}\rho_\infty = 0$. The fact that $A$ is an eigenoperator and not just a symmetry with $[H, A] = 0$ is crucial and guarantees that the operators $\rho_{nm}$ are not stationary for $n \neq m$. We refer to these $\rho_{nm}$ as mixed coherences because they describe oscillations induced by $\mathfrak{H}$ between, usually mixed, stationary states $\rho_{nm}$ [see Fig. 1b]. In contrast to the coherences in a decoherence-free subspace they are not decoupled from the environment and are affected by dissipation $L_\mu \rho_{nm} L_\mu^\dagger \neq 0$. All initial states that contain mixed coherences $\rho_{nm}$ will continuously oscillate in the long-time limit. If only one operator $A$ exists then the spectrum of the dark Hamiltonian is equidistant, like that of a harmonic oscillator. The equidistance of the spectrum ensures the long-time dynamics is

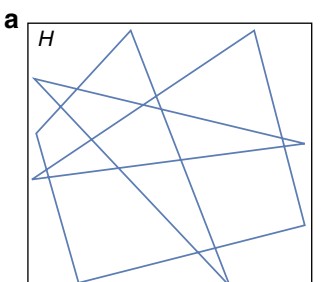
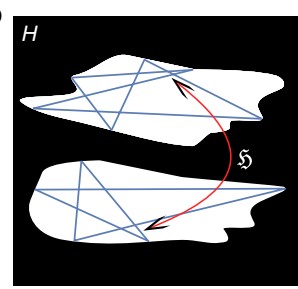

**Fig. 1** The Hilbert space $\mathcal{H}$ separated into a decaying part (black) and a nondecaying part (white). **a** Ergodic time evolution, indicated by a blue trajectory, explores the entire connected nondecaying space thus leading to stationarity of observables after a transient period. **b** Dissipation may split the nondecaying part into disjoint sectors. The dark Hamiltonian $\mathfrak{H}$ drives transitions between them. Observables that are not entirely contained in one of these parts show continued oscillations and thus nonergodic behavior after a transient period

periodic, with period $2\pi/\lambda$ and the system does not relax to stationarity.

**Nonstationary dynamics in the open Hubbard Model.** We study the emergence of nonstationarity in $D$-dimensional fermionic Hubbard models that possess spin and $\eta$-pairing symmetries (see Methods). This is a paradigmatic example that can be accurately realized in highly controllable quantum systems, such as optical lattices filled by ultracold spin 1/2 atoms[21]. The Hamiltonian is

$$H = -\tau \sum_{\langle j,j'\rangle,s} c_{j,s}^\dagger c_{j',s} + c_{j',s}^\dagger c_{j,s} + \sum_j U n_{j,\uparrow} n_{j,\downarrow} + \epsilon_j n_j + \frac{B}{2}\left(n_{j,\uparrow} - n_{j,\downarrow}\right),$$
(3)

where $j, j'$ denotes nearest-neighbor sites of a bipartite lattice with $M$ sites and $c_{j,s}$ is the annihilation operator for a fermion with spin $s$ on site $j$. The particle number operator is $n_{j,s} = c_{j,s}^\dagger c_{j,s}$ and $n_j = n_{j,\uparrow} + n_{j,\downarrow}$. The hopping amplitude is $\tau$, $U$ denotes onsite interactions and $\epsilon_j$ is a site dependent energy offset. In an optical lattice, the term $\epsilon_j$ describes the trapping potential and/or spin-agnostic disorder, e.g., created through speckle patterns[3,22]. A constant external magnetic field splits different spin states by $B$ via the Zeeman effect. We assume the coupling of the Hubbard lattice to the environment to take the form of local dephasing Lindblad operators $L_j = \gamma_j n_j$. In optical lattices, this can be achieved e.g., through immersion into a Bose–Einstein condensate[23] (see Supplementary Discussion for details).

The strong symmetries[17,19,24] of this model determine its generalized grand-canonical-like equilibrium states as $\rho_\infty \propto \exp\left(\beta_0 N + \beta_1 (S^+ S^-) + \beta_2 S^z\right)$, where $N$ is the total number of particles and $\mathbf{S} = (S^x, S^y, S^z)$ the total spin (note that the stationary subspace is degenerate). The parameters $\beta_i$ play the role of generalized chemical potentials determined by the initial state. The operator $S^+$ fulfills the criteria of the eigenoperator $A$ with $\lambda = B$ and hence constructs a dark Hamiltonian $\mathfrak{H}$ (see Methods).

We study the system evolution starting from noncorrelated polarized initial states. In Fig. 2a we show the bulk-averaged fermion spin along the $x$-direction $\langle\langle S_i^x(t)\rangle\rangle$. The long-time oscillation amplitude of spin–spin correlations $\langle S_i^x(t) S_{i+j}^x(t)\rangle$ for arbitrary $i$ and $j$ are shown in Fig. 2b. After a short-transient time these observables start oscillating with an amplitude quickly converging to a finite value with increasing $M$. Their spectrum is then narrowly centered around multiples of $B$ as shown in Fig. 2c. This is in excellent agreement with the analytically expected purely sinusoidal evolution in the long-time limit. In Fig. 2d we compare traces of the spin dynamics in the $xy$-plane for different initial spin polarizations. All realizations (see Methods) of the stochastic dynamics are identical for the maximally polarized state, which thus behaves similarly to an isolated collection of noninteracting spins. However, realizations for nonmaximally polarized states possess fluctuations that increase with system size $M$. Only after averaging many realizations perfectly sinusoidal oscillations emerge following the initial transient. This evolution strongly violates ergodicity and is qualitatively different from the precession of independent spins.

In Fig. 3a, we study a quench starting from the ground state of the Hubbard model. In the absence of dephasing, the combination of disorder and many-body thermalization quickly dampens out the dynamics, as shown in the inset of Fig. 3a. The closed system exhibits small fluctuations following revivals due to finite-size effects. Remarkably, in the presence of dephasing[22,23] persistent spin oscillations with frequency $B$ ensue after the quench. The strength of the system environment coupling solely determines the time for the transient dynamics to decay and coherent, oscillatory behavior appear in the measured observables. In Fig. 3b, we show that fundamentally quantum off-diagonal long-range order[25] in the spin sector $\lim_{n\to\infty}\langle S_i^+ S_j^-\rangle \neq 0, \forall i, j$ is constructed by the dephasing dynamics even when starting from high-temperature thermal states of the Hubbard model.

We emphasize that the eigenstates of the dark Hamiltonian $\mathfrak{H}$ driving these oscillations are mixed and cannot be realized in an isolated system. Furthermore, the system admits no decoherence-free subspaces as any state $|\phi\rangle$ for which $L_j|\phi\rangle = 0, \forall j$ cannot be an eigenstate of $H$ for finite hopping $\tau$. Indeed, all coherences that lead to dephasing in the isolated system get damped out by the dissipation because the setup does not admit dark states (see Supplementary Methods for a more detailed discussion).

We apply well-established complexity measures based on entropy[10] (see Methods) to the time evolution induced by $\mathfrak{H}$. Figure 4a shows the mutual information between lattice sites as a function of time. In the presence of dephasing we see that for small times this is uniform and large which indicates that the reduced quantum state of a single site contains a large amount of information about the rest of the system. During the time evolution the mutual information decreases while simultaneously the disparity, shown in Fig. 4b, increases. Even a relatively small system reaches a complex state with little mutual information and large disparity between different sites. This is consistent with Fig. 2d showing large fluctuations in individual realizations of the evolution. The experimental characterization of such a state necessarily requires measuring many sites. Figure 4 also shows that this complexity does not emerge in the closed system.

## Discussion

Starting from the Hubbard model, different couplings to the environment can realize different classes of dark Hamiltonians (see the Supplementary Discussion for the details). For instance, when $\epsilon_j = \epsilon$ spin dephasing $L_j = \gamma_j S_j^z$ results in a dark Hamiltonian whose eigenstates all possess long-range off-diagonal $\eta$-pairing order, i.e., $\lim_{n\to\infty}\langle \eta_i^+ \eta_j^-\rangle \neq 0, \forall i, j$. Spin dephasing could thus contribute to the formation of superconducting states by inducing $\eta$-pairing[26].

More generally, our results open up the possibility of studying quantum statistical physics[27] of non-Hermitian[28] dark Hamiltonians. Linear response theory, behavior under periodic driving, relaxation toward subspaces of the dark Hamiltonian, the formulation of a semiclassical limit and metastability[29] are also interesting and open questions. The asymptotic coherent dynamics induced by a dark Hamiltonian breaks time-translation symmetry. It may thus be understood as the dissipative realization of a fully quantum time crystal[30,31] in the bulk that does not require external time-dependent driving[31] or collective dissipation of a noninteracting system[32].

We have shown that relaxation to equilibrium and stationarity can be prevented by environmental dissipation. This causes some degrees of freedom to dampen out and stops them from dephasing. The underlying physics resembles classical complex system dynamics where also not all available degrees of freedom contribute to the formation of collective complex behavior[9].

## Methods

**Symmetries of the Hubbard model.** The $D$-dimensional Hubbard Hamiltonian on a bipartite lattice commutes with two sets of generators of the $su(2)$ algebra. The

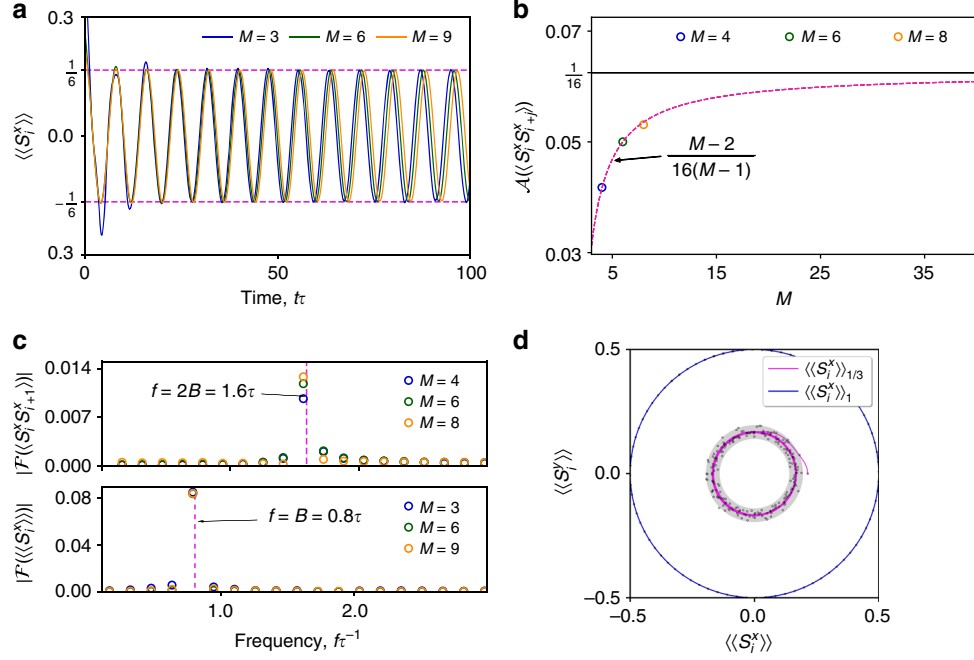

**Fig. 2** Dynamics of spin observables in the number-dephased Hubbard model following a quench. **a** The bulk-averaged fermion spin $\langle\langle S_i^x(t)\rangle\rangle$ for various system sizes $M$. The evolution starts from the half-filled lattice state $\langle\langle S_i^x\rangle\rangle_{1/3}$ without double-occupancies where every third fermion is polarized along $-x$ while all others are polarized along $x$. A long-time amplitude of $1/6$ independent of $M$ is obtained analytically. **b** Amplitude of the oscillations of $\langle S_i^x(t)S_{i+j}^x(t)\rangle$ in the long-time limit for different $M$ and arbitrary $i, j$. The starting state is the maximally polarized quarter filled state with a fermion put on every second site. The magenta dashed curve shows the analytical result converging to $1/16$ in the limit $M \to \infty$. **c** The spectra obtained from the dynamical evolution in **a** and **b** for times $t \in [20, 100]/\tau$ are strongly peaked around multiples of $B$ as expected in the long-time limit. **d** Traces of the polarization in the $xy$-plane starting from the maximally polarized starting state $\langle\langle S_i^x\rangle\rangle_1$ (blue curve) and from the state $\langle\langle S_i^x\rangle\rangle_{1/3}$ (magenta curve) and $M = 9$. The solid lines are averages over 2000 trajectories (see Methods). The markers show values from a single realization and the shaded area indicates the range of typical fluctuations of a realization. All calculations were carried out for $B = 0.8\tau$, $U = \tau$ and $\gamma = 0.4\sqrt{\tau}$ without disorder $\epsilon_j = 0$

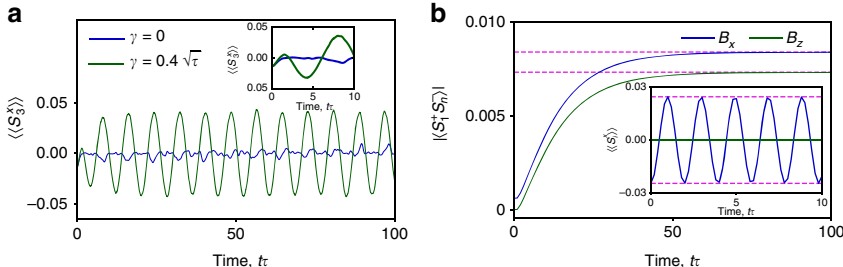

**Fig. 3** Time evolution of spin observables in both the open and closed Hubbard model during a quench. **a** Dynamics of $\langle\langle S_3^x(t)\rangle\rangle$ for the open and the closed system following a quench (the inset shows relaxation and the first revival of the finite-size closed system). The initial state is the Hubbard model ground state with $N = 5$ particles, $M = 7$ sites, $U = \sqrt{2}\tau$, $B = 0$ and no disorder. At time $t = 0$ the system is quenched to $U = \tau$, $B = 0.8\tau$ and disorder $\epsilon_j \in [0.0, 0.16]\tau$. **b** Dynamics of $\left|\langle S_1^+(t)S_n^-(t)\rangle\right|$ starting from a high-temperature thermal state $\propto \exp(-\beta H)$ with $\beta = 0.2/\tau$, $U = \tau$ and $B = 0.2\tau$ of $N = 4$ fermions in $M = 5$ sites. The magnetic field initially points either along the $x$-direction (blue lines) or the $z$-direction (green lines). At time $t = 0$ dephasing $\gamma = 0.4\sqrt{\tau}$ is switched on and $B$ pointed along the $z$-direction. Long-range correlations emerge from the initially thermal state through dephasing. The inset shows $\langle\langle S_i^x(t)\rangle\rangle$ and the magenta dashed lines represent analytical values in the long-time limit

first set consists of spin operators[33]

$$S^z = \sum_j S_j^z, \quad S_j^z = \tfrac{1}{2}(n_{j,\uparrow} - n_{j,\downarrow}), \tag{4}$$

$$S^+ = \sum_j S_j^+, \quad S_j^+ = c_{j,\uparrow}^\dagger c_{j,\downarrow}, \tag{5}$$

$$S^- = \sum_j S_j^-, \quad S_j^- = c_{j,\downarrow}^\dagger c_{j,\uparrow}, \tag{6}$$

where $c_{j,\downarrow}$ ($c_{j,\uparrow}$) is the standard fermionic annihilation operator annihilating a down (up) spin on site $j$. We have

$$[H, S^z] = 0, \quad [H, S^\pm] = \pm B\, S^\pm. \tag{7}$$

The other, hidden, $SU(2)$ symmetry, called $\eta$-pairing, is given in terms of its generators as

$$\eta^z = \tfrac{1}{2}\sum_j (n_j - 1), \tag{8}$$

$$\eta^+ = \sum_j \tau(j)\eta_j^+, \quad \eta_j^+ = c_{j,\uparrow}^\dagger c_{j,\downarrow}^\dagger, \tag{9}$$

$$\eta^- = \sum_j \tau(j)\eta_j^-, \quad \eta_j^- = c_{j,\downarrow} c_{j,\uparrow}, \tag{10}$$

where $\tau(j)$ follows an alternating checkerboard pattern of $+1$. With $\epsilon_j = \epsilon$ we have

$$[H, \eta^z] = 0, \quad [H, \eta^\pm] = \pm 2\epsilon\, \eta^\pm. \tag{11}$$

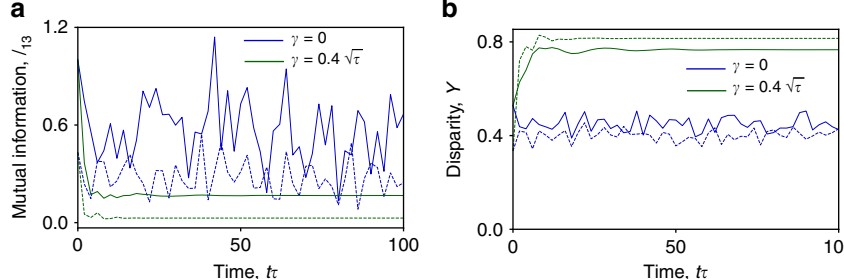

**Fig. 4** Measures of complexity applied to the evolution of a random initial state of the open and closed Hubbard model. The data is for $N = 3$ particles on $M = 4$ sites (dotted lines) and $N = 4$ particles on $M = 4$ sites with the first, second, and fourth sites being polarized along the $x$-direction and the third site along $-x$ (solid lines). **a** The quantum mutual information between sites 1 and 3 that decreases quickly during the initial part of the evolution and is noticeably less for the open system case. **b** The average disparity $Y$ between lattice sites growing with time for the open system dynamics and remaining approximately constant in the closed system. Other parameters as in Fig. 2

Crucially, we also have $\left[ S_j^\alpha, \eta_k^\beta \right] = 0, \forall \alpha, \beta, j, k$. This fact allows us to construct Lindblad operators in terms of either spin or $\eta$-pairing operators and get dark Hamiltonians in the long-time limit. In the main text we study the example $L_j = \gamma_j n_j$. Explicitly, the local transverse magnetizations are given by $S_j^x = (S_j^+ + S_j^-)/2$ and $S_j^y = i(S_j^+ - S_j^-)/2$.

**Quantum mutual information and disparity**. Taking a complex network measure applied to quantum systems from ref. [10] we study the complexity of the coherent dynamics using quantum mutual information

$$I_{ij} = \tfrac{1}{2}\left( \mathcal{S}_i + \mathcal{S}_j - \mathcal{S}_{ij} \right), \qquad (12)$$

where $\mathcal{S}_i = \mathrm{tr}(\rho_i \log \rho_i)$ and $\mathcal{S}_{ij} = \mathrm{tr}(\rho_{ij} \log \rho_{ij})$ are the one- and two-point reduced von Neumann entropies of subsystems $\rho_i = \mathrm{tr}_{k \neq i} \rho$ and $\rho_{i,j} = \mathrm{tr}_{k \neq i,j} \rho$. Using this we also define the disparity $Y_i$

$$Y_i = \frac{\sum_{j=1}^M (I_{ij})^2}{\left( \sum_{j=1}^M I_{ij} \right)^2}, \qquad (13)$$

which may intuitively be understood by observing that it is small when the quantum mutual information between site $i$ and the other sites takes on a constant value and large when one particular $I_{ij}$ takes on a dominant value. More specifically, we study the average disparity across the sites $Y = \frac{1}{M} \sum_{j=1}^M Y_j$.

**Simulation of the master equation**. The numerical calculations shown in Figs. 2 and 3a were performed by a stochastic unraveling of the master equation into individual realizations by the quantum trajectories method[34]. The trajectories were calculated using the Tensor Network Theory Library[35]. In Figs. 3b and 4, we numerically integrated the full matrix representation of the master equation directly.

## Data availability

The data that supports the plots within this paper and other findings of this study are available from the authors upon reasonable request. The figures were produced with Python and processed with Inkscape.

## Code availability

The Tensor Network Theory Library[35], which can be used to perform the simulations in the article, is available at http://www.tensornetworktheory.org/. The programming scripts used to obtain the data in this manuscript are available from the authors upon reasonable request.

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

## Acknowledgements

We thank A. Buchleitner, L. Carr, C. Foot, G. L. Giorgi, E. Ilievski, M. Kiffner, J. Mur-Petit, T. Prosen, U. Schneider, and R. Smith for useful discussions. We also thank A. Lazarides for suggesting studying the effects of disorder in our setup. The work has been supported by EPSRC grants No. EP/P009565/1 and EP/K038311/1 and is partially funded by the European Research Council under the European Union's Seventh Framework Program (FP7/2007-2013)/ERC Grant agreement no. 319286 Q-MAC. We acknowledge the use of the University of Oxford Advanced Research Computing (ARC) facility in carrying out this work (https://doi.org/10.5281/zenodo.22558).

## Author contributions

B.B. identified the conditions for nonstationarity and carried out the analytical calculations. J.T. performed the numerical calculations. D.J., B.B and J.T. discussed and analyzed the results and their physical implications and contributed significantly to writing the manuscript.

## Additional information

**Competing interests:** The authors declare no competing interests.

