## [Peer Review File · Nature Communications]

Reviewers' comments:

Reviewer #1 (Remarks to the Author):

Dear Editor,

The authors satisfactorily answered all my questions and made appropriate corrections in the manuscript. The only remaining point on which we disagree is calling the dark Hamiltonian non-Hermitian, (as they also explain) this generator is Hermitian under a proper identification of the space on which it acts. This is a minor point, and in the new version, it is also not featured prominently.

I recommend that the article is published in its current version.

Martin Fraas

Reviewer #3 (Remarks to the Author):

The authors present a concept of time-dependent decoherence free subspaces and apply it to driven open quantum many body systems, to show in several examples based on spin models and the fermionic Hubbard model in one dimension that as a physical consequence, certain observables show persistent oscillations. This goes beyond the previously established concept of dark states in driven open many-body systems, where the decoherence free subspace is typically time independent (one reason being that it is typically one-dimensional, such that any time evolution would be trivial).

I find this paper truly inspiring and I have no doubt that the conclusions are correct, since all general arguments are complemented with concrete numerical simulations in one-dimensional systems. In particular, I judge the step beyond the previous dark state scenario an important one; one reason being that it may lead to a rather robust scenario that may be ubiquitous in Hamiltonian quantum systems with strong symmetries, once coupled to (quite natural) dissipation channels, as illustrated in the Hubbard model example (but see my questions below). In this light, if my concerns below can be addressed, I would recommend publication in Nature Communications.

Referee 1 raises the question of originality of the results, pointing at several papers likewise concerned with time-dependent decoherence free subspaces. I went through some of these references (which I was likely not aware of) and based on that I agree that the mathematical structure reported by the authors was recognized previously. However, in all of these works the focus is on small quantum systems, instead of many-body systems. The examples given there appear rather artificial, if present at all. In the 'worst case', the present work may be seen as leveraging these works to the many-body realm, and recognizing that the dynamical symmetries needed for this setting to work are actually realized in several interesting and relevant many-body Hamiltonians, in terms giving these symmetries even a new physical meaning.

In summary, I think that the proper contextualization sometimes may just as valuable as the pure mathematical construction, and this is the actual advance of the present work. Of course, this is a personal view, but there are many examples in physics (e.g. the Berry phase was discovered in optics way before Berry found it, but he figured out the broad scope and applicability of the

concept).

Obviously, I still suggest that the authors properly account for the previous work.

Here are some more specific comments and concerns, which revolve around the robustness of their effect in the many-body context (so answers are important to strengthen my own reasoning on novelty and contextualization):

1. Realization of the fermion model: Why is it necessary to immerse the fermion system into a bath of bosons? Is the spontaneous emission from the lattice drive laser not sufficient to induce local dephasing? I.e. should one use less far detuned lattices? This is an ubiquitous effect present in any optical lattice (see H. Pichler, A. J. Daley, and P. Zoller, Phys. Rev. A 82, 063605 (2010)): So, why have experiments not seen the coherent oscillations so far?
2. More generally: What is the requirement on the strength of dephasing: Does it have to be much larger than the other scales in the Hamiltonian? Does one need a gap (minimal distance from the imaginary axis in their Supplementary Fig. 1? (Which I do not see, upon increasing the lattice size).
3. How big are the decoherence free subspaces in the context of the one-dimensional examples for thermodynamically large systems? The background of this question is whether, in case they are extensively large (number of states contains scales with system size -- seems compatible with their sub- n^2 scaling?), then there should be thermalization/ergodic dynamics be going on in either the subspace. Would that overwrite their effect or is it robust under such subspace ergodicity?
4. In the presence of dephasing (more generally: hermitian jump operators), there is always the fully mixed state stationary solution, which is annihilated by the jump operators and commutes with any Hamiltonian. Usually, this fixed point is attractive in a large system. The author however propose a generalized Gibbs ensemble as the stationary solution. Why, in this light? What about the unit matrix stationary solution?
5. Physical picture: While I follow their mathematical construction, I stumbled over the following (seeming?) contradiction, which the authors could help clarify and improve their physical picture (honestly, I find the high-level description of the effect starting in line 45 rather confusing than enlightening; in particular, what means "symmetry protecting"? symmetry preserving?): If I understand it correctly, there are stationary states in the problem (termed ρ_∞), but certain observables show persistent oscillations. Usually, one would argue that, given a stationary state density matrix, any correlation function evaluated on it is stationary as well (depends only on the difference in times between the observables, e.g. $\langle \psi^\dagger(t) \psi(t') \rangle$ is a function of $t-t'$ alone). In their case, there is a forward in time evolution for some observables despite a stationary state. How is this understood more intuitively? Or conversely, what is the meaning of the stationary state ρ_∞ in this context?
6. Line 121, what means " $n \rightarrow \infty$ " in this context? Obviously the filling/density is bounded.

Reply to Referees of "Non-stationary coherent quantum many-body dynamics through dissipation: Revision 1"

B. Buca, J. Tindall, D. Jaksch

1 Reply to Referee 1

Referee 1:

"The authors satisfactory answered all my questions and made appropriate corrections in the manuscript. The only remaining point on which we disagree is calling the dark Hamiltonian non-Hermitian, (as they also explain) this generator is Hermitian under a proper identification of the space on which it acts. This is a minor point, and in the new version, it is also not featured prominently.

I recommend that the article is published in its current version."

We thank the referee again for their very useful and interesting remarks, suggestions and questions, as well as their positive appraisal of our manuscript and the recommendation to publish our article.

2 Reply to Referee 3

We thank the referee for their interesting questions and useful comments. We address the concerns raised on a point-by-point basis.

Referee 3:

"The authors present a concept of time-dependent decoherence free subspaces and apply it to driven open quantum many body systems, to show in several examples based on spin models and the fermionic Hubbard model in one dimension that as a physical consequence, certain observables show persistent oscillations. This goes beyond the previously established concept of dark states in driven open many-body systems, where the decoherence free subspace is typically time independent (one reason being that it is typically one-dimensional, such that any time evolution would be trivial).

I find this paper truly inspiring and I have no doubt that the conclusions are correct, since all general arguments are complemented with concrete numerical simulations in one-dimensional systems. In particular, I judge the step beyond the previous dark state scenario an important one; one reason being that it may lead to a rather robust scenario that may be ubiquitous in Hamiltonian quantum systems with strong symmetries, once coupled to

(quite natural) dissipation channels, as illustrated in the Hubbard model example (but see my questions below). In this light, if my concerns below can be addressed, I would recommend publication in Nature Communications.

Referee 1 raises the question of originality of the results, pointing at several papers likewise concerned with time-dependent decoherence free subspaces. I went through some of these references (which I was likely not aware of) and based on that I agree that the mathematical structure reported by the authors was recognized previously. However, in all of these works the focus is on small quantum systems, instead of many-body systems. The examples given there appear rather artificial, if present at all. In the “worst case”, the present work may be seen as leveraging these works to the many-body realm, and recognizing that the dynamical symmetries needed for this setting to work are actually realized in several interesting and relevant many-body Hamiltonians, in terms giving these symmetries even a new physical meaning. In summary, I think that the proper contextualization sometimes may just as valuable as the pure mathematical construction, and this is the actual advance of the present work. Of course, this is a personal view, but there are many examples in physics (e.g. the Berry phase was discovered in optics way before Berry found it, but he figured out the broad scope and applicability of the concept). Obviously, I still suggest that the authors properly account for the previous work.”

We thank the referee for their positive appraisal of the manuscript. We now include citations to all the previous work mentioned by both of the referees. We were also not aware of some of the work mentioned. We now address the points raised.

1. *Referee 3:*

” 1. Realization of the fermion model: Why is it necessary to immerse the fermion system into a bath of bosons? Is the spontaneous emission from the lattice drive laser not sufficient to induce local dephasing? I.e. should one use less far detuned lattices? This is an ubiquitous effect present in any optical lattice (see H. Pichler, A. J. Daley, and P. Zoller, Phys. Rev. A 82, 063605 (2010)): So, why have experiments not seen the coherent oscillations so far? ”

In the limit of red-detuning, provided we can neglect the off-diagonal terms in the dissipator, the dominant effect of the scattering is to return the atoms to the lowest band, and the corresponding effect of the lattice dissipation is to introduce pure local on-site dephasing (c.f. eq. (28) of the reference) as stated by the referee.

The spin-1/2 fermions in our ultra-cold atom setting are implemented as two atomic hyperfine levels. The incoherent light scattering will, in general, distinguish between the two spins. This would result in two Lindblad operators per site $L_{i,\uparrow} = n_{i,\uparrow}$ and $L_{i,\downarrow} = n_{i,\downarrow}$, where $n_{i,\uparrow}$ is the number of spin-up, spin-down fermions on site i . In order to respect the condition below line 74 of the main text of our manuscript, we require that the Lindblad operators commute with the total spin raising operator. This might be achievable in some experiments by fine-tuning the lattice laser parameters but will in general not be the case.

In contrast, in the setup described in the Supplementary Material the BEC interacts via spin-independent density-density interaction. This guarantees that the decoherence will not distinguish between spin-up and down. This difference is subtle, but crucial. Physically, the first case corresponds to a measurement (classical random process) that distinguishes between the different spin states. The second case corresponds to a measurement in a spin-agnostic way.

2. *Referee 3:*

" 2. More generally: What is the requirement on the strength of dephasing: Does it have to be much larger than the other scales in the Hamiltonian? Does one need a gap (minimal distance from the imaginary axis in their Supplementary Fig. 1? (Which I do not see, upon increasing the lattice size)."

There is no requirement on the strength of the dephasing within our model. We only need that other sources of dissipation that do not respect the symmetry requirement are small compared to it. What happens when there are additional large sources of such unwanted dissipation is an interesting open question. We plan to study this in the near future - one may imagine the possibility of a dissipative phase transition depending on the ratio of the strengths of the different types of dissipation. This question is related to the one of metastability that we have mentioned in the concluding paragraphs.

The strength of the dissipation does control the size of the gap and thus the time it takes for the system to reach the non-stationary infinite-time limit. We have added a sentence to the last paragraph of p. 3 to make this clearer: "The strength of the system environment coupling determines the time for the transient dynamics to decay and coherent, oscillatory behaviour to appear."

Supplementary Fig. 1 shows the spectrum of the XXZ spin ring with a single loss term. With increasing system size, the gap closes and new frequencies enter into the long-time dynamics. If these are dense and incommensurate, eigenstate dephasing (in the closed system sense) is possible and this could happen for the XXZ spin ring. An analytical and/or numerical analysis of this thermodynamic limit is an interesting open question. We note that in contrast to the XXZ spin ring we do not see new frequencies entering the long-time oscillatory dynamics in the Hubbard model discussed in the main text.

We now comment on the possibility of eigenstate dephasing in the XXZ spin ring in the Supplementary Material by including the sentence: "With increasing system size, new frequencies enter into the long-time dynamics. If these are dense and incommensurate, eigenstate dephasing does become a possibility."

3. *Referee 3:*

"3. How big are the decoherence free subspaces in the context of the one-dimensional examples for thermodynamically large systems? The background of this question is whether, in case they are extensively large (number of states contains scales with system size – seems compatible with their $sub - n^2$ scaling?), then there should be

thermalization/ergodic dynamics be going on in either the subspace. Would that overwrite their effect or is it robust under such subspace ergodicity?"

We estimate the size of the subspaces for the Hubbard model example is $M^2 - M$. There can be thermalization/ergodic dynamics within each subspace - that is the crucial difference between this example and a usual decoherence free subspace composed of pure states. Even when each individual subspace has relaxed by itself (mathematically, the eigenspaces contain only mixed states), remarkably, the joint dynamics will still be non-stationary. This is because the subspaces all have commensurate frequencies and thus they can never dephase each other in the closed system sense (destructive interference due to incommensurate frequencies).

In order to make it clearer that the effect is robust regardless of the thermalization in each individual subspace we have changed the sentence on p. 3 of the main text that said: "The long-time dynamics is then periodic with period $2\pi/\lambda$ " to: "The equidistance of the spectrum ensures the long-time dynamics is periodic, with period $2\pi/\lambda$ and the system does not relax to stationarity."

4. *Referee 3:*

"4. In the presence of dephasing (more generally: hermitian jump operators), there is always the fully mixed state stationary solution, which is annihilated by the jump operators and commutes with any Hamiltonian. Usually, this fixed point is attractive in a large system. The author however propose a generalized Gibbs ensemble as the stationary solution. Why, in this light? What about the unit matrix stationary solution?"

The referee is correct. The unit matrix is a stationary state which corresponds to setting $\beta_0 = \beta_1 = \beta_2 = 0$ in our parametrization of stationary states. The important point is that this is not the only stationary point in our system. The symmetry properties discussed in the manuscript require the quantities $N, S^Z, S^+ S^-$ to be conserved during the evolution. This in general prevents the dynamics from reaching the unit matrix stationary state.

To emphasize the importance of these quantities being conserved we have added the following sentence: "The quantities $N, S^Z, S^+ S^-$ are conserved during the time evolution."

5. *Referee 3:*

"5. Physical picture: While I follow their mathematical construction, I stumbled over the following (seeming?) contradiction, which the authors could help clarify and improve their physical picture (honestly, I find the high-level description of the effect starting in line 45 rather confusing than enlightening; in particular, what means "symmetry protecting"? symmetry preserving?): If I understand it correctly, there are stationary states in the problem (termed ρ_∞), but certain observables show persistent oscillations. Usually, one would argue that, given a stationary state density matrix, any correlation function evaluated on it is stationary as well (depends only on the difference in times between the observables, e.g. $\langle \psi^\dagger(t) \psi(t') \rangle$ is a function of $t-t'$ alone). In their case, there is a forward in time evolution for some observables despite a stationary state. How is this understood more intuitively? Or conversely, what is the meaning of the stationary

state ρ_∞ in this context?”

We agree with the referee that the term “symmetry protecting” is confusing and have changed the wording to “symmetry preserving”. This phrase makes the symmetric nature of the dissipation clearer.

The referee is correct, our system features multiple stationary states ρ_{mm} . Importantly we also find eigenmodes ρ_{nm} with $n \neq m$ that have purely imaginary eigenvalues and hence do not damp out in the long-time limit. If these eigenmodes are contained in the initial state the system will never reach a stationary state. Instead, the eigenmodes with purely imaginary eigenvalues will continuously oscillate. They are the physical origin of the oscillations in single-time observables seen in our work. The two-time correlation functions mentioned by the referee may also show interesting behaviour, which we will study in the future.

We realize that our notation ρ_∞ may have given the wrong impression that this is the only state that can be reached after a long time. In order to rectify this we have now unified the notation of eigenmodes and renamed $\rho_\infty \rightarrow \rho_{00}$.

6. *Referee 3:*
" 6. Line 121, what means " $n \rightarrow \infty$ " in this context? Obviously the filling/density is bounded."

We thank the referee for pointing this typo. We have removed it and replaced it with $\forall i, j$.

3 List of changes

1. We have added a sentence to the last paragraph of p. 3 to make this clearer: "The strength of the system environment coupling determines the time for the transient dynamics to decay and coherent, oscillatory behaviour to appear."
2. We have added two sentences to supplementary material at the end of the 'Dynamical decoherence-free subspace: The XXZ spin ring' subsection: "With increasing system size, new frequencies enter into the long-time dynamics. If these are dense and incommensurate, eigenstate dephasing does become a possibility."
3. We have changed the sentence on p. 3 of the main text that said: "The long-time dynamics is then periodic with period $2\pi/\lambda$ " to: "The equidistance of the spectrum ensures the long-time dynamics is periodic, with period $2\pi/\lambda$ and the system does not relax to stationarity."
4. We have added the following sentence in par. 3 of p. 3 of the main text: "The quantities $N, S^Z, S^+ S^-$ are conserved during the time evolution."

5. On line 45 in the main text we have change the term "symmetry protecting" to "symmetry-preserving".
6. In cases when the symmetry conditions from the main text are satisfied we have renamed $\rho_\infty \rightarrow \rho_{00}$ throughout the main text and the Supplementary Material. To provide for notational consistency in the Supplementary Material below Cor. 1 on p. 6. we added a sentence: "For cases corresponding to Cor. 1 we rename $\rho_\infty \rightarrow \rho_{00}$ ".
7. Fixed typo in the equation on line 121 (removed $n \rightarrow \infty$ and put $\forall i, j$).

REVIEWERS' COMMENTS:

Reviewer #3 (Remarks to the Author):

The authors have carefully addressed my questions and concerns. Some questions had to be left open as topics of future research, but this just emphasizes that this is an intriguing and potentially quite rich topic.

I recommend publication of the manuscript in the present form.